# Haemodynamic Instability and Brain Injury in Neonates Exposed to Hypoxia–Ischaemia

**DOI:** 10.3390/brainsci9030049

**Published:** 2019-02-27

**Authors:** Shiraz Badurdeen, Calum Roberts, Douglas Blank, Suzanne Miller, Vanesa Stojanovska, Peter Davis, Stuart Hooper, Graeme Polglase

**Affiliations:** 1The Ritchie Centre, Hudson Institute of Medical Research, Monash University, Melbourne 3168, Australia; calum.roberts@monash.edu (C.R.); Douglas.Blank@monashhealth.org (D.B.); suzie.miller@monash.edu (S.M.); vanesa.stojanovska@hudson.org.au(V.S.); stuart.hooper@monash.edu (S.H.); graeme.polglase@monash.edu (G.P.); 2Newborn Research Centre, The Royal Women’s Hospital, Melbourne 3052, Australia; pgd@unimelb.edu.au

**Keywords:** hypoxic–ischaemic encephalopathy, asphyxia, neonate, cord clamping, oxygen, haemodynamic

## Abstract

Brain injury in the asphyxic newborn infant may be exacerbated by delayed restoration of cardiac output and oxygen delivery. With increasing severity of asphyxia, cerebral autoregulatory responses are compromised. Further brain injury may occur in association with high arterial pressures and cerebral blood flows following the restoration of cardiac output. Initial resuscitation aims to rapidly restore cardiac output and oxygenation whilst mitigating the impact of impaired cerebral autoregulation. Recent animal studies have indicated that the current standard practice of immediate umbilical cord clamping prior to resuscitation may exacerbate injury. Resuscitation prior to umbilical cord clamping confers several haemodynamic advantages. In particular, it retains the low-resistance placental circuit that mitigates the rebound hypertension and cerebrovascular injury. Prolonged cerebral hypoxia–ischaemia is likely to contribute to further perinatal brain injury, while, at the same time, tissue hyperoxia is associated with oxidative stress. Efforts to monitor and target cerebral flow and oxygen kinetics, for example, using near-infrared spectroscopy, are currently being evaluated and may facilitate development of novel resuscitation approaches.

## 1. The Problem

Birth asphyxia is the term used to describe the continuum of physiological derangements seen in newborn infants as a result of a prolonged or profound mismatch between oxygen demand and oxygen delivery. It ranges from mild to severe; however, when severe, it can cause irreversible cerebral cell damage and death, leading to a syndrome of hypoxic–ischaemic encephalopathy (HIE). HIE may lead to altered conscious state, autonomic instability, absence of primitive reflexes, seizures, and death. Worldwide, birth asphyxia accounts for 24% of neonatal deaths and around 800,000 deaths annually under the age of five years [1]. For survivors, the burden of long-term morbidity at the level of individuals, families, and society is immense. While therapeutic hypothermia is used clinically to reduce neurological injury secondary to HIE, there remains a 45–55% risk of death or moderate–severe disability in treated infants [2,3,4]. There is, therefore, the need to explore other means of secondary prevention aimed at diminishing the burden of a major contributor to death and developmental impairment in term infants.

## 2. A Window for Reducing Further Brain Injury

Whole-body and selective head cooling target the secondary phase of reperfusion injury and energy failure, and must be commenced within six hours of birth [5]. An important window for reducing brain injury exists in the delivery room prior to the commencement of therapeutic hypothermia. A better understanding of the changes in haemodynamics and oxygen kinetics in the first few minutes after birth provides an opportunity to improve resuscitation strategies. Delivery room management of asphyxic infants currently focuses on the rapid achievement of lung aeration, cardiac output, and oxygenation. Resuscitation with air has been shown to reduce mortality compared to resuscitation using 100% oxygen, highlighting the importance of avoiding oxygen toxicity [6]. Beyond this, for the most compromised infants, there is little clinical evidence to guide resuscitation [7,8]. In this review, we detail evidence of impaired cerebral autoregulation that renders brain tissue vulnerable to haemodynamic instability and untargeted oxygen delivery during the initial transition period after birth. We summarise evidence showing that early umbilical cord clamping contributes to haemodynamic instability and brain injury, and we outline possible brain-protective resuscitation strategies to improve the outcomes of asphyxic newborns. 

## 3. Physiological Changes During Asphyxia

Birth asphyxia may result from a variety of perinatal events, including placental detachment, placental blood flow disruption, foetal blood loss, compression of the umbilical cord, prolonged labour, maternal hypoxia, and failure of the newborn to initiate pulmonary gas exchange [9]. All of these result in reduced circulating blood oxygen levels of differing severity. Whether or not they result in a reduced oxygen delivery to end-organ tissues depends on the tissue. It is unsurprising that, given its dependent position *in utero*, the foetus has very efficient in-built compensatory mechanisms that allow it to survive with reduced oxygen delivery. Experiments in sheep showed that the foetus responds to hypoxia by ceasing breathing (central primary apnoea) and body movements and by redistributing cardiac output. While, initially, foetal hypoxia induces a vagal bradycardia, this changes into a sustained tachycardia (within an hour) as the hypoxia continues [10]. These changes reduce metabolic demand in the face of impaired oxygen delivery and increase blood flow to vital organs such as the heart and brain to maintain oxygen delivery (DO_2_), where cerebral DO_2_ = cerebral blood flow × blood oxygen content. 

The foetal response to continuing asphyxia diverges, depending upon its severity and whether the umbilical cord remains intact. If survivable, the redistribution of cardiac output is sustained eventually (weeks/months), leading to an asymmetric growth restriction, with a significantly higher brain-to-body/weight ratio. Interestingly, despite the sustained hypoxemia, foetal breathing movements and body movements return, indicating a resetting of the chemoreceptor-induced inhibition of breathing to a much lower oxygenation level [11]. The impact of this on spontaneous breathing in growth-restricted infants is unknown. However, if the cause of the asphyxia is severe, leading to a continued or acute reduction in oxygenation, primary apnoea is followed by a period of irregular gasping driven by primitive spinal centres [12]. A foetus born in this condition is likely to need more intensive resuscitation (Figure 1).

### 3.1. Myocardial Function

As the duration of asphyxia increases, these adaptive mechanisms are overwhelmed by a continued reduction in oxygen levels, which, despite the increase in coronary blood flow, leads to a reduction in cardiac oxygen delivery. As myocardial metabolism is highly oxygen-dependent, the reduction in oxygen delivery combined with the gradual exhaustion of tissue oxygen stores (myoglobin) leads to a loss in myocardial function [13]. As a result, blood pressure gradually decreases as myocardial function fails (Figure 1), eventually resulting in asystole. If an infant is delivered in this state (or near asystole), restoration of myocardial function is critical to restoring cardiac output and oxygen delivery to essential end-organs such as the brain. Since the loss in myocardial function is primary due to a loss in myocardial oxidative metabolism, logically, resuscitation should be focused on re-supplying the heart with oxygen. Recent animal experiments demonstrated that, although lung aeration is critical to facilitate pulmonary gas exchange and to dilate the pulmonary circulation, chest compressions and adrenaline are also required to facilitate flow in the coronary circulation [14,15]. To restore spontaneous circulation (myocardial function), current guidelines recommend ventilation with 100% oxygen, assuming that it will more rapidly replenish myocardial oxygen supply. However, in animal experiments, ventilation with air is equally effective and reduces the risk of rebound hyperoxia. Ventilation with a gas mixture containing 5% oxygen was only marginally slower at restoring heart rate and blood pressure, whereas ventilation with 100% nitrogen prevented the restoration in cardiac function with ventilation onset [16]. The latter demonstrates the importance of restoring oxidative metabolism in this process. 

### 3.2. Cerebral Circulation

In response to asphyxia, autoregulatory signalling pathways at the endothelial–astrocyte junctions maintain relatively constant cerebral blood flow despite falling oxygen levels [17]. As asphyxia progresses and oxygen delivery continues to fall, mitochondrial bioenergetic failure and depletion of ATP and phosphocreatine ensues. Anaerobic metabolism leads to lactate accumulation, decoupling of autoregulatory mechanisms, and compromise of the blood–brain barrier [18,19]. The maximally dilated vasculature renders the cerebral circulation pressure passive [20]. Recent studies in asphyxic-term-equivalent lambs, irrespective of whether they needed chest compressions or adrenaline, showed that the current practice of cord clamping prior to ventilation results in rebound hypertension that rapidly follows the return of spontaneous circulation (ROSC) [15,21]. This rebound hypertension was associated with markers of cerebral injury, including blood vessel protein extravasation and loss of tight-junction integrity in the subcortical and periventricular white matter and cortical grey matter. As the cerebral vascular resistance vessels are maximally dilated, the delicate microvasculature is exposed to markedly elevated pressures, increasing the risk of damage. Other studies in non-asphyxic preterm lambs showed that umbilical cord clamping prior to ventilation results in a rapid increase in cerebral blood flow, systolic pressure, and cerebral oxygen extraction [22,23]. This hypertension is the result of both increased post-ductal vascular resistance, following removal of the placental bed, and chemoreceptor-mediated sympathetic activation with increasing time between umbilical cord clamping and ventilation [24]. 

## 4. Failure of Oxygen Delivery and the Provision of Supplementary Oxygen

In utero, the preductal arterial oxygen saturation (SaO_2_) of predominantly foetal haemoglobin in normal foetuses is approximately 60% [25,26]. During brief foetal distress, the arterial oxyhaemoglobin saturation measured by pulse-oximetry (SpO_2_) may decrease to 30% (corresponding to a pO_2_ of 12–15 mmHg) or even lower, with no significant acidosis [27,28]. Following an uncomplicated birth, the SpO_2_ in normal infants rapidly rises to reach on average 80% by 3–4 min and >90% by 6–7 min [29]. This rise corresponds with lung aeration and a rapid decrease in pulmonary vascular resistance. Failure of SpO_2_ to increase may signal inadequate lung aeration and/or elevated pulmonary vascular resistance and a persistence of the foetal circulation. An increase in SpO_2_ is necessary to avoid persistent pulmonary hypertension and acidosis in light of increasing metabolic demand caused by breathing and thermogenesis [30]. 

During transition of the non-asphyxic infant, animal experiments showed that lung aeration is the predominant factor driving the increase in pulmonary blood flow, with oxygen playing a supplementary role [16,31]. However, as indicated above, in severely asphyxic infants, restoring myocardial oxygen supply also requires an increase in coronary blood flow. Current neonatal resuscitation guidelines recommend supplementary oxygen if the infant does not respond to initial lung aeration and requires chest compressions [7,8]. However, there are scarce data to guide the use of supplementary oxygen during resuscitation and after ROSC to prevent or minimise further hypoxic brain injury [32,33]. The asphyxic infant appears to be especially at risk of oxidative stress-mediated brain injury. High oxygen concentrations lead to an increase in reactive oxygen species which cause both microvascular and parenchymal injury. Damage is exacerbated by impaired autoregulation and increased cerebral blood flow [16,34]. Currently, guidelines suggest titrating oxygen therapy to achieve similar saturations to those seen in normal-term infants in the first minutes after birth. It is recommended that infants requiring cardiac compressions for severe bradycardia (heart rate <60 beats per minute) receive 100% oxygen. It is possible that a more restrictive approach to oxygen therapy may reduce the burden of oxygen toxicity and, as indicated above, air was shown to be equally effective as 100% O_2_ in animal studies. The approach of starting with 21% oxygen and titrating therapy based on oxygen saturation in severely asphyxic infants requiring cardiac compressions remains to be studied. Following successful resuscitation and restoration of cardiac output and oxygen delivery to tissues, the rapid increase in cardiac output and oxygen delivery causes a secondary reperfusion injury. This includes disturbances of intracellular homeostasis, neuronal excitotoxicity, and free-radical-mediated injury. It occurs over a period of at least 6–24 hours and is mitigated by therapeutic hypothermia [35].

## 5. A Physiology-Based Resuscitation Approach

Initial resuscitation should aim to achieve the following:Minimise ischaemic time by maintaining cardiac output and oxygenation until lung aeration is established;Avoid reperfusion injury by targeting a graded normalisation of cerebral perfusion pressure and oxygen kinetics (Figure 2).

The foetus is reliant upon umbilical venous flow, flowing through the ductus venosus then directed across the patent foramen ovale, for up to 50% of left ventricular preload and left ventricular cardiac output [36,37]. Following lung aeration, pulmonary resistance falls dramatically. Left-heart filling switches from umbilical venous flow to pulmonary venous flow [38]. It follows, therefore, that maintaining umbilical venous return until after lung aeration and the increase in pulmonary circulation occurs is integral to maintaining systemic cardiac output. Current guidelines recommend clamping the umbilical cord prior to the commencement of ventilation. Experiments conducted in non-asphyxic preterm lambs demonstrated that delaying umbilical cord clamping until after the establishment of ventilation prevents the dramatic drop in heart rate seen following immediate cord clamping with delayed ventilation [36]. Umbilical cord clamping prior to ventilation causes a drop in venous return and cardiac output of about 50% within 60 seconds and, with increasing cord clamping-to-ventilation intervals, there is progressive hypoxia, asphyxia, and bradycardia [21,36]. A further advantage of a physiological-based umbilical cord-clamping approach is likely to be the maintenance of a source of oxygenated blood from the placenta until the lungs begin to assume the role of gas exchange. These two major advantages are most likely to benefit (i) the mildly–moderately asphyxic infant, where cardiac contractility is still intact and placental circulation is not completely compromised, and (ii) infants where there is a delay between cord clamping and successful lung aeration, for example, in low-resource settings [39,40]. The maintenance of venous return to the left heart is critical for the hypoxic newborn. The redistribution of cardiac output is essential for maintaining adequate oxygen delivery to important organs such as the brain, heart, and adrenals, at the expense of less important organs (skin, gut, muscles). Immediate cord clamping abolishes this protective mechanism by compromising cardiac output [21,36], making the hypoxic newborn more prone to significant brain injury. However, in a model of severely asphyxic lambs, heart rate, cardiac output, and oxygen saturation levels were profoundly depressed in both immediate and physiological-based cord-clamping groups [15]. This is likely to reflect grossly compromised oxygenation and cardiac contractility in the setting of established hypoxia–ischaemia. 

Exposure of the cerebral vascular bed to fluctuations in blood pressure and flow following restoration of cardiac output may be equally important for the asphyxic infant [15,23,24]. It appears that clamping the umbilical cord after ventilation is established is only half the picture. Delaying cord clamping in term-equivalent asphyxic lambs until well after ROSC greatly mitigated the rebound hypertension and brain injury. This was likely due to the presence of the highly compliant, low-resistance placental vascular bed within the systemic circuit during the recovery in cardiac function which acted as a “pressure-relief valve” for the upper body circulation. Interestingly, despite a >20-mmHg difference in arterial pressure at 5 min following ventilation onset, carotid arterial blood flow was similar between immediate and delayed cord-clamping groups. This reflects a marked change in the pressure/flow relationship, perhaps because the cerebral vascular bed was maximally dilated, and cerebral blood flow was at a maximum in both groups [15]. In addition, studies in preterm lambs showed that umbilical cord clamping prior to ventilation results in a decrease in arterial oxygen saturation and cerebral oxygenation [22]. By providing continued return of oxygenated blood to the left side of the heart, delayed cord clamping improved cerebral oxygenation—an effect that correlated with improved systemic oxygenation and cerebral perfusion [15,22,36].

### Placental Transfusion

Delayed cord clamping in preterm and vigorous-term infants has been shown to increase haemoglobin concentrations and reduce the need for blood transfusions [41,42]. Whether or not this results from a “placental transfusion” is a matter of debate. Nevertheless, there appears to be a net transfer of haemoglobin from the placental reservoir to the infant after birth that is dependent not just on time, but on multiple physiological factors [38]. Foremost amongst these is the change in pulmonary vascular resistance and ductal flow following lung aeration. Other factors may include spontaneous breathing, gravity, uterine contractions, and umbilical arterial spasm. During asphyxia, the physiological response of redistributing cardiac output away from non-vital organs increases systemic vascular resistance, which contributes to a greater proportion of cardiac output perfusing the lower resistance placenta; this is an adaptive response for the foetus. Both *in utero* and intrapartum asphyxia cause net shifts in placental and foetal blood volume, which may also be affected by the duration and cause of asphyxia [43,44]. A delay in umbilical cord clamping until after the infant’s systemic vascular resistance normalises may allow redistribution of cardiac output back into the foetal peripheral circulation, away from the placenta, thereby increasing the infant’s blood volume. This concept is yet to be demonstrated in animal or human experiments. Immediate umbilical cord clamping in asphyxic infants prior to resuscitation may contribute to the lower haemoglobin levels shortly after birth compared to healthy newborns, although this finding may be related to antepartum haemorrhage being the aetiology of asphyxia in some infants [45].

Whether or not allowing placental “transfusion” to take place by delaying umbilical cord clamping following an asphyxial injury leads to better infant outcomes is unclear. On the one hand, infants with asphyxia have impaired cardiac output that may be mitigated by improved preload, and increased haemoglobin is likely to improve oxygen carrying capacity. However, this may exacerbate reperfusion injury from reactive oxygen species. Increased blood viscosity may compromise organ perfusion at the level of capillary beds [46]. A retrospective cohort study showed higher mortality but better long-term outcomes amongst survivors in anaemic asphyxic infants compared to non-anaemic asphyxic infants [47]. This difference may, however, be related more to differences in the aetiology of asphyxia and response to therapeutic hypothermia between the two groups. Indeed, these differences highlight the need for personalising resuscitation strategies.

## 6. Personalised Resuscitation

While the range of clinical parameters that may be used for monitoring in the delivery room is growing, it remains unclear which might help the clinician mitigate brain injury. Currently, reasonable guidance exists on how to best commence resuscitation in both preterm and full-term neonates. Measurement of novel cardiorespiratory parameters may allow individualised therapy throughout transition. Progress in resuscitation care will depend on moving away from broad assumptions and “one size fits all” strategies. 

One attractive option is to avoid exposing the injured brain to excessive surges in perfusion. This first requires accurate determination of ROSC following successful resuscitation. At present, ROSC is determined by auscultation of the infant’s heart rate. The use of electrocardiography is becoming increasingly popular in the delivery room, but there is an important risk of mistaking pulseless electrical activity for a return of heart rate and cardiac output [48]. End-tidal carbon dioxide capnography is now recommended adult practice both to monitor cardiopulmonary resuscitation quality and to aid identification of ROSC [49]. A recent experiment in piglets corroborated these findings in a model of neonatal asphyxia [50]. 

Following ROSC, cerebral oxygen delivery (including blood flow) may be carefully titrated to avoid secondary injury. Components of oxygen delivery that may be manipulated in the delivery room are oxygen saturation and cerebral vascular dilatation via control of partial pressure of carbon dioxide (pCO_2_). Crucially, avoidance of a rapid rise in blood pressure during transition is likely to be desirable. This may be achieved by keeping the placenta attached to the systemic circulation until well after the sympathetic and catecholamine-driven surge in cardiac output plateaus. However, it remains unclear when the ideal time is to clamp the cord. There may be a role for monitoring preductal blood pressure in the delivery room. Currently available methods utilise the inflation of a cuff and may even provide continuous blood pressure monitoring [51], but these interrupt the SpO_2_ signal and may not be acceptable for clinical use during resuscitation. Another possibility that is currently available in some pulse oximeters is monitoring of the “perfusion index”. The perfusion index is the ratio of the pulsatile blood flow to the non-pulsatile or static blood in peripheral tissue, thus representing a non-invasive measure of systemic vascular resistance [52]. A rise in the perfusion index may indicate relaxation of peripheral vasoconstriction during reperfusion, allowing the placental circulation to be safely removed from the circulation. 

Monitoring peripheral arterial oxygen saturation provides little information about the adequacy of oxygen delivery, as it does not account for cerebral blood flow, haemoglobin concentration, or metabolic demand. Prolonged cerebral hypoxia–ischaemia is likely to contribute to perinatal cerebral injury, while hyperoxia is associated with oxidative stress [53]. There is increasing interest in continuous monitoring of cerebral regional tissue oxygen saturation (crSO_2_) and fractional tissue oxygen extraction (FTOE) using near-infrared spectroscopy (NIRS) during the foetal-to-neonatal transition [54,55,56]. Cerebral tissue oxygenation reaches a plateau faster than peripheral arterial oxygen saturation, indicating preferential oxygen delivery to the brain in the first minutes after birth [57,58]. In theory, a rapid rise in crSO_2_ and/or high absolute level may be a useful marker for an undesirable surge in oxygenation/perfusion to the brain. In preterm lambs, crSO_2_ was shown to correlate well with cerebral blood flow in conditions of constant arterial oxygen saturation and metabolic demand. Concordance is best in the time domain, where crSO_2_ correlates well across a moderate range of flow volume changes that are allowed to reach a steady state. When blood flow fluctuates, crSO_2_ corresponds with low-frequency variations with greater fidelity, and is likely to reflect the time delay in achieving equilibrium of oxygenation in the venous blood pool [59,60].

Interpreting crSO_2_ may be challenging in the context of acute recovery from an asphyxial insult, as multiple variables are likely to be rapidly changing. These include SpO_2_, metabolic demand, pCO_2_, and haemoglobin (from delayed cord clamping), ductal and foramen ovale shunt direction, and, therefore, left ventricular output [61,62]. These factors will all potentially influence crSO_2_ readings, but may be concurrently monitored and accounted for by calculating FTOE (FTOE = (SpO_2_ − crSO_2_)/SpO_2_), metabolic demand by monitoring amplitude integrated electroencephalography (aEEG) [63], and end-tidal CO_2_ as a surrogate for PaCO_2_ [64,65]. An approach that goes beyond manipulating supplementary oxygen levels in response to crSO_2_ and includes a consideration of these multiple parameters seems physiologically logical, but may require automated feedback algorithms and is yet to be evaluated [55,66,67]. For example, a drop in crSO_2_ may reflect a drop in PaCO_2_ from overventilation and subsequent reduction in cerebral blood flow, and would be best addressed by reducing minute ventilation rather than increasing the fraction of inspired oxygen. A limitation, however, is that changes in crSO_2_ are likely to lag behind these rapidly changing physiological variables, but may be reflective of slower changes in oxygen extraction and consumption at the tissue level. Nevertheless, the core principle stands, in that the pattern of change in crSO_2_, when measured reliably, provides a global picture of (mis)match between oxygen delivery (which includes blood flow) and metabolic demand. The piglet brain exposed to hypoxia–ischaemia is being used to develop computational models of alterations in blood flow and metabolism that can also help to explain changes at a cellular level [68]. NIRS-derived changes in cerebral blood volume were recently used to determine the optimal blood pressure that supports autoregulatory function during therapeutic hypothermia; blood-pressure deviation from optimal autoregulatory vasoreactivity was associated with magnetic resonance imaging markers of brain injury that were independent of the initial birth asphyxia [69]. It is yet to be demonstrated that NIRS monitoring can individualize blood pressure, SpO_2_, and pCO_2_ targets to reduce ischemia–reperfusion injury in the delivery room. 

## 7. Limitations and Practical Considerations

Most of the knowledge derived to date is from a mix of preterm and term animal studies. It is unknown how similar or different their responses may be in relation to cerebral autoregulation in the setting of perinatal asphyxia in humans. Studies in neonates are challenging because antenatal monitoring is often non-specific in identifying which infants will be born with signs of asphyxia. Research in the emergency setting of resuscitation of an infant who was not anticipated to be asphyxic at birth will require novel trial methodology and tailored ethical considerations. 

The benefit of physiologically-based cord clamping is likely to be dependent on the cause of asphyxia. In cases of placental or cord compromise, delaying cord clamping may provide little or no benefit in terms of continued venous return to the neonate. On the other hand, leaving the placental circulation intact may confer protection from rebound hypertension. How long the placental circulation should be left intact is still unknown but may be directed by non-invasive markers that indicate a normalisation in blood pressure for the individual neonate. Consideration will also need to be given to the active management of third stage of labour.

Several groups have developed techniques and equipment to facilitate neonatal resuscitation with cord intact. These range from high-end, custom-built devices (Concord, Lifestart) [70,71] to more low-cost, low-tech approaches (VentFirst pole, BabyDUCC) [72,73], and even a platform that may be suitable in low-resource settings [74]. Of note, the Concord system facilitates the monitoring of multiple physiological variables during resuscitation through the incorporation of a respiratory function monitor that may allow targeting of physiological goals during transition. Feasibility studies have demonstrated that resuscitation can successfully be provided at the mother’s side with the cord intact, and efficacy studies are under way. While there are currently no studies evaluating this resuscitation approach specifically for infants with birth asphyxia, it is the natural next step if the physiological benefits are confirmed.

## 8. Conclusions

Haemodynamic instability during delivery room management of asphyxic infants is increasingly being recognised and may represent a therapeutic target with potential for secondary prevention of injury. Resuscitation with the placental circulation intact is feasible and may attenuate the cerebrovascular hypertension that evolves during recovery from the acute hypoxic insult. Future studies should address both prolonged oxygen deficiency and oxygen-mediated toxicity in the delivery room, and more work is needed to define what parameters are most useful to monitor in this regard.

## Figures and Tables

**Figure 1 brainsci-09-00049-f001:**
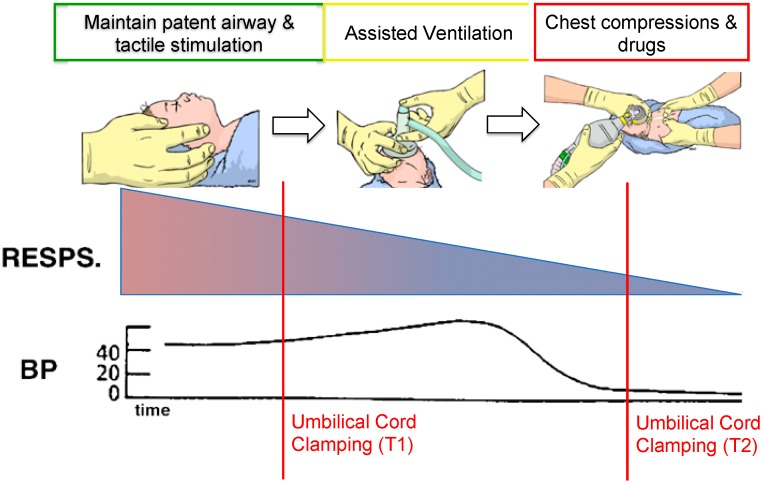
Cardiorespiratory changes during asphyxia. As *in utero* hypoxia–ischaemia progresses, compensatory physiological mechanisms are overwhelmed, and the infant requires more intensive resuscitation. The vertical red lines correspond to examples of times of birth coupled with immediate cord clamping as per current practice. Immediate cord clamping may contribute to a further drop in heart rate at timepoint T1 if achievement of lung aeration is delayed, or it may impair restoration of cardiac output at timepoint T2. Adapted from the Neonatal Resuscitation Programme (NRP) Neonatal Resuscitation Textbook 6th Edition (2011), American Academy of Pediatrics, and American Heart Association [12].

**Figure 2 brainsci-09-00049-f002:**
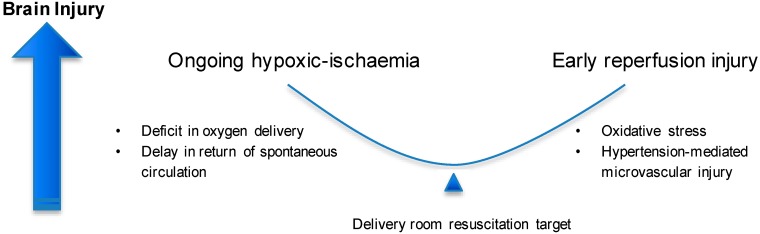
Factors contributing to ongoing brain injury during resuscitation of the infant at risk of asphyxia. The aim of delivery room management should be to balance, on the one hand, hypoxia–ischaemia duration, with exposure of cerebral tissue to excessive oxygen delivery and blood pressure.

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
