# Peer review of "Haemodynamic Instability and Brain Injury in Neonates Exposed to Hypoxia–Ischaemia"

_brainsci, 2019, doi:10.3390/brainsci9030049_

Round 1

Reviewer 1 Report

The manuscript provides a review of principles underlying brain injury in the asphyxiated newborn infant and draws upon current physiological research to suggest novel approaches to minimizing perinatal brain injury.  This well-written review effectively links translational physiology and explores potential new clinical pathways to minimize asphyxial injury.

1.                   The Problem – At the end of this section, there is an opportunity to even more explicitly state the challenge/goal of reducing hypoxic-ischemic injury through other means of secondary prevention and thus diminish the burden of a major contributor to developmental impairment in term infants.

2.                   A window for reducing further brain injury – This section nicely lays out the particulars, but would be even stronger after a sweeping statement of scope.

3.                   Physiological changes during asphyxia – Is there a better descriptor for the fetus’ physiologic position in utero as an alternative to “subjugate”? My dictionary says “bring under domination or control, especially by conquest”.  Maybe a less charged term would be “subordinate” or “dependent”.

3.2 Line 112 – Should this read “….in asphyxiated term-equivalent lambs, irrespective of whether they needed…..”

4.            Failure of oxygen delivery and the provision of supplementary oxygen – In line 138, “Therefore” does not make an obvious link with the previous sentence referring to an increase in coronary blood flow.

5.            A physiology-based resuscitation approach – In line 187, might provide a reference for the statement, “Immediate cord clamping abolishes this protective mechanism” (referring to redistribution of cardiac output to maintain oxygen delivery to critical organs).

                In this section “asphyxiated” and “asphyxic” appear to be used as equivalent terms.  Would it be best to maintain consistent usage of one or the other term?

                In line 201, the between group comparison being described is unclear.  Could the comparison groups be stated explicitly?

                5.1  In this section is it important to reference the early work of Linderkamp and Yao which suggested that intrapartum asphyxia promotes transfer of blood volume to the fetus (Linderkamp O et al. Eur J Paediatr 1978; 127:91)?

6.            Personalized resuscitation – Would it be worthwhile discussing improved cardiac monitoring in the delivery room for more accurate quantitation of heart rate and determination of ROSC?  Devices such as the NeoBeat dry electrode device offer the potential to obtain a heart rate within seconds after birth.  This might limit unnecessary chest compressions, allowing more appropriate focus on ventilation and transition from placental to pulmonary function.

                In line 269 “concordance is best in the time domain across a moderate range of flow volume changes” has unclear meaning.  Can this be re-phrased?

                In line 292 “….markers of brain injury that were…..”         

7.            Line 300 “….an infant who was not anticipated….”  Babies are persons who merit the personal pronoun “who”.

8.            Conclusions – This is another opportunity to emphasize the big picture “…a therapeutic target with potential for secondary prevention of asphyxia injury.”

References – References are current and accurately cited.  There is some errant capitalization in references 25 and 27.

Reviewer 2 Report

The review by Badurdeen and colleagues titled “Haemodynamic Instability and Brain Injury in Neonates Exposed to Hypoxia-Ischaemia” is very well written and very relevant summary of an important issue involving neonatal hypoxia-ischemia. This is a thorough and easy to follow review.

Minor comments:

The authors mention under the limitations section that a vast majority of work is performed in a mix of in vivo models, which may not directly translate to human cases. In line with this very accurate statement, the authors should take care and not discuss animal model and human cases simultaneously, so that the reader is clear on where the information comes from. For example, in the section 3, it is assumed that the authors are discussed newborns, as per the text and references 9, 11 and 12. However, reference 10 refers to the fetal sheep model. Therefore, it would be best if the authors made the distinction between clinical and animal model research clear throughout the text.

The authors mention throughout the manuscript animal models of neonatal hypoxia-ischemia, however, the authors focus mostly on the fetal sheep model. There is a need to at least comment on what is known in the piglet model of term hypoxia-ischemia. Even though it is a postnatal model, and therefore, clamping of the umbilical cord is not commonly part of the experimental routine, the piglet model is a well-established translational model, with research also conducted on cerebral haemodynamics following hypoxia-ischemia.
